# Biochemical markers after the Norseman Extreme Triathlon

**Christoffer Nyborg**[ID][1,2]*, **Jørgen Melau**[ID][1,2,3], **Martin Bonnevie-Svendsen**[2], **Maria Mathiasen**[4], **Helene Støle Melsom**[1,2], **Andreas B. Storsve**[ID][5], **Jonny Hisdal**[1,2]

**1** Faculty of medicine, Department of Clinical medicine, University of Oslo, Oslo, Norway, **2** Section of Vascular Investigations, Oslo University Hospital, Aker, Norway, **3** Prehospital Division, Vestfold Hospital Trust, Tønsberg, Norway, **4** Department of Cardiology, Telemark Hospital Trust, Notodden, Norway, **5** Aker BioMarine Antarctic AS, Lysaker, Norway

* nyborgchristoffer@gmail.com

**Data Availability Statement:** All relevant data are within the manuscript and its Supporting Information files. Rawdata for age and race times are not included to ensure anonymity for the

## Abstract

Prolonged exercise is known to cause changes in common biomarkers. Occasionally, competition athletes need medical assistance and hospitalisation during prolonged exercise events. To aid clinicians treating patients and medical teams in such events we have studied common biomarkers after at The Norseman Xtreme Triathlon (Norseman), an Ironman distance triathlon with an accumulated climb of 5200 m, and an Olympic triathlon for comparison. Blood samples were collected before, immediately after, and the day following the Norseman Xtreme Triatlon (n = 98) and Oslo Olympic Triathlon (n = 15). Increased levels of clinical significance were seen at the finish line of the Norseman in white blood cells count (WBC) (14.2 [13.5–14.9] 109/L, p < 0.001), creatinine kinase (CK) (2450 [1620–3950] U/L, p < 0.001) and NT-proBNP (576 [331–856] ng/L, p < 0.001). The following day there were clinically significant changes in CRP (39 [27–56] mg/L, p < 0.001) and Aspartate Aminotransferase (AST) (142 [99–191] U/L, p < 0.001). In comparison, after the Olympic triathlon distance, there were statistically significant, but less clinically important, changes in WBC (7.8 [6.7–9.6] 109/L, p < 0.001), CK (303 [182–393] U/L, p < 0.001) and NT-proBNP (77 [49–88] ng/L, p < 0.01) immediately after the race, and in CRP (2 [1–3] mg/L, p < 0.001) and AST (31 [26–41] U/L, p < 0.01) the following day. Subclinical changes were also observed in Hemoglobin, Thrombocytes, K+, Ca2+, Mg2+, Creatinine, Alanine Aminotransferase and Thyroxine after the Norseman. In conclusion, there were significant changes in biomarkers used in a clinical setting after the Norseman. Of largest clinical importance were clinically significant increased WBC, CRP, AST, CK and NT-proBNP after the Norseman. This is important to be aware of when athletes engaging in prolonged exercise events receive medical assistance or are hospitalised.

## Introduction

There is a growing interest in ultra-triathlons competitions worldwide [1, 2]. Occasionally, competition athletes need medical assistance and hospitalisation due to fatigue, illness or

participants since the results from the races are public.

**Funding:** This study was founded through a grant form Aker BioMarine Antarctic AS (http://www.akerbiomarine.com/). The funder provided support in the form of salaries for author ABS and research materials but did not have any additional role in the study design, data collection and analysis, decision to publish, or preparation of the manuscript. The specific roles of the author are articulated in the 'author contributions' section."

**Competing interests:** The funder provided support in the form of research materials and salaries for author ABS but did not have any additional role in the study design, data collection and analysis, decision to publish, or preparation of the manuscript. This does not alter our adherence to PLOS ONE policies on sharing data and materials.

accidents [3]. Prolonged exercise is known to cause changes in common biomarkers [4–6]. There are major changes in serval biomarkers after Ironman distance triathlons [7–9]. Knowledge about what range of change can be considered "normal" regarding the most common biomarkers used in clinical diagnostics is therefore warranted, as this will allow for the findings in patients who have participated in these types of competitions to be evaluated [10, 11]. Our aim is to publish a reference material for medical teams working at The Norseman Xtreme Triathlon (Norseman) and other Ironman triathlons.

Norseman has been rated as one of the world's toughest triathlons by popular media [12]. It is probably one of the most popular Xtreme Triathlon Competition in the world, with more than 5000 applicants for the start slots in the 2019 race. It is also the current host of the World Championship in triathlon. The competition is an Ironman distance triathlon that takes place in Norway [13]. It starts with a jump from a ferry and a 3800 meter swim in the Hardanger Fjord. Then the athletes race 180 km by bicycle with an elevation of more than 3000 meters, before finishing off with a 42.2 km run to Gaustatoppen. The total elevation of the race course is 5200 m.

We examined Creatinine Kinase (CK), N-terminal pro Brain Natriuretic Peptide (NT-proBNP), Creatinine, C-reactive protein (CRP), White blood cells (WBC), Thrombocytes, Hemoglobin (Hb), Sodium (Na$^+$), Potassium (K$^+$), Calcium (Ca$^{2+}$), Magnesium (Mg$^{2+}$), Aspartate Aminotransferase (AST), Alanine Aminotransferase (ALT), Triiodothyronine (T3), Thyroxine (T4) and Thyroid Stimulating Hormone (TSH), covering heart function, inflammation markers, markers of cell injury, electrolytes and thyroid function [4–9, 14–20]. For comparison of the impact of different types of triathlons we have also examined the same biomarkers after an Olympic triathlon. Our hypothesis where that there would be greater changes I the biomarkers after Norseman compared to after an Olympic triathlon.

## Materials and methods

### Study population

In the present study, volunteers were recruited from among participants in the Norseman races of 2016, 2018 and 2019, and from the Oslo Triathlon (Olympic distance) in 2018. Norseman and Oslo Triathlon are both contested in Norway in August. All participants received an e-mail 1–2 months prior to the races with information and the possibility to sign up for the study. All volunteers that completed the respective races were included. All volunteers signed an informed consent form before being included in the study. The study was approved by the Regional Committees for Medical and Health Research Ethics in Norway (Protocol number: REK Sør-Øst 2016/932) and performed in accordance with the declaration of Helsinki.

### Blood sampling

Blood samples were collected from an antecubital vein within 24 hours prior to the start of each race (Baseline), as well as immediately after finish (Finish), and at noon of the following day (Day after). Bioengineers and medical doctors were present at all times during testing, both for the collection and handling of the samples. Whole blood samples were drawn on K$_2$EDTA (ethylene diamine tetra acetic acid) vacutainers and refrigerated immediately after blood sampling. Serum samples were drawn in serum vacutainers containing silica particles and gel separators. These were then clotted at room temperature for 30 min before being centrifuged at 2000 g for 10 min and then refrigerated. All tubes were transported refrigerated to a certified clinical laboratory (Fürst medisinsk laboratorium, Oslo, Norway) for analysis. Only complete sets of three samples were included for analysis. Analysis were conducted of CK,

NT-proBNP, Creatinine, CRP, WBC, Thrombocytes, Hb, $Na^+$, $K^+$, $Ca^{2+}$, $Mg^{2+}$, AST, ALT, T3, T4 and TSH.

## Data management and statistics

All data handling, statistics and plotting was performed in R [21]. Normality for all measurements was tested with the Shapiro-Wilk Normality Test. As only CRP had normal distributed values for all measurements, all data is presented and tested with non-parametric methods to ease the reading of the article, while simultaneously maintaining the statistical strength of the findings.

Statistical tests were performed on paired data with Wilcoxon Signed Rank Tests and unpaired data with Wilcoxon Rank Sum tests to assess changes from baseline measurements. The results are given as (median [1. Quartile, 3. Quartile], p-value). Correlations were performed with Spearman's rank correlation and presented as (R-value, p-value). Correlations were performed on all possible paired values for each correlation. Critical alpha level was set as 0.05 for all statistical tests.

## Results

### Samples

In total 139 subjects registered and completed baseline measurements. Due to a lack of follow up, a total of 113 subjects were included for analysis with full sets of samples (Baseline, Finish, Day after): 38 samples from the Norseman 2016, 28 from the Norseman 2018, 31 from the Norseman 2019 and 15 from the Oslo Triathlon 2018. Characteristics of included subjects are given in Table 1. Due to accident some vials was damaged during transport and excluded from the analysis. Therefore, there are differences in the total number of analyses performed for each biomarker. Complete numbers of analysis per biomarker per year are given in Appendix 1.

### Blood sample analysis

Clinically significant increased levels were seen at the finish line of the Norseman in WBC (12.7 [11.1–15.9] $10^9$/L, p < 0.001), CK (2450 [1620–3950] U/L, p < 0.001) and NT-proBNP (576 [331–856] ng/L, p < 0.001). CRP had an initial small increase at the finish line after the

**Table 1. Characteristics of included subjects.**

| | Norseman | | Olympic distance | |
|---|---|---|---|---|
| | **Female** | **Male** | **Female** | **Male** |
| n | 26 | 72 | 3 | 12 |
| Age (Y) | 38 [34–42] | 34 [34–48] | 39 [33–41] | 43 [37–47] |
| Height (cm) | 168 [165–170] | 180 [176–184] | 168 [164–176] | 182 [177–187] |
| Mass (kg) | 61 [57–64] | 78 [71–84] | 61 [58–66] | 73 [72–81] |
| BMI (m/kg$^2$) | 21.5 [20.2–22.5] | 24.1 [22.8–25.7] | 21.1 [21.0–21.4] | 22.3 [21.0–23.8] |
| **Times** | | | | |
| Swim (min) | 76 [64–92] | 76 [68–84] | 33 [33–34] | 32 [26–33] |
| Bike (min) | 447 [410–507] | 418 [390–460] | 85 [77–88] | 73 [69–78] |
| Run (min) | 360 [327–394] | 350 [315–385] | 54 [50–56] | 47 [44–51] |
| Total (min) | 902 [828–996] | 874 [793–930] | 180 [167–182] | 158 [143–169] |

Values are median [1. Quartile, 3. Quartile]

Norseman (8 [4–19] mg/L, p < 0.001) with a clinically significant increase the following day (39 [27–56] mg/L, p < 0.001 compared to baseline, < 0.001 compared to finish line). AST had increased at the finish line (99 [74–136] U/L, p < 0.001) but it increased to even higher values the day after the race (142 [99–191] U/L, p < 0.001 compared to baseline, p < 0.05 compared to finish line). CK had significantly increased at the finish line compared to baseline, and continued to increase. It displayed the highest values the day after the races (2910 [1650–4730] U/L, p < 0.001 compared to baseline).

In comparison, after the Olympic triathlon there were also statistically significant changes, though of less clinical importance, in WBC (7.8 [6.7–9.6] $10^9$/L, p < 0.001), CK (303 [182–393] U/L, p < 0.001) and NT-proBNP (77 [49–88] ng/L, p < 0.01) at the finish line and for CRP (2 [1–3] mg/L, p < 0.001) and AST (31 [26–41] U/L, p < 0.01) the day following the event. There were significantly lower elevations in the above measurements after the Olympic triathlon compared with values after the Norseman competitions (all p < 0.001).

We conducted a correlation analysis of the clinically significant blood samples after the Norseman races. We examined correlations among the variables, reported weekly exercise, race time, age, and body mass index (BMI). NT-proBNP at the finish line was negatively correlated with BMI (-0.35, p < 0.001). There were no significant correlations for WBC. AST and CRP the day after, and CK at the finish line, were positively correlated with each other. AST was positively correlated with CRP (0.27, p < 0.05), CK (0.82, p < 0.001), and age (0.28, p < 0.05). CRP was positively correlated with AST [0.27, p < 0.05], CK (0.43, p < 0.001), BMI (0.26, p < 0.05) and race time (0.33, p < 0.01). CK was positively correlated with AST (0.82, p < 0.001), CRP (0.43, p < 0.001), BMI (0.34, p < 0.001) and race time (0.26, p < 0.01). A visualised correlation matrix is given in Fig 1.

Subclinical changes were also observed in Hb, Thrombocytes, $K^+$, $Ca^{2+}$, $Mg^{2+}$, Creatinine, AST, ALT and T4 after the Norseman and for $K^+$, $Mg^{2+}$, Creatinine, AST and TSH after the Olympic distance. All results with statistical comparison to baseline values are given in Table 2. The percentage of samples out of reference range is given in Table 3 and individual values for WBC, CRP, AST, CK and NT-proBNP are illustrated in Fig 2. The single highest value seen in CRP, AST, ALT and CK is from the same subject and the individual was found clinically well after the race and at a control 2 weeks after the race.

## Discussion

The main finding in the present study is that the majority of biomarkers used in clinical evaluation increase above reference values after prolonged exercise events, such as the Norseman. The changes are generally much more significant after the Norseman compared to the Olympic distance. The findings of greatest clinical interest are those in WBC, CRP, AST, CK and NT-proBNP because the results would indicate pathology in the resting state in a medical examination [22–26]. This is in line with previous studies on Ironman distance triathlons [7–9].

A correlation analysis of our data shows that the increase in WBC and NT-proBNP seems to be separate phenomena, while the increase in AST and CK is strongly correlated and shows some correlation with the increase in CRP.

### WBC

Measurements of WBC show that prolonged exercise is related to leukocytosis, which declined the day after the Norseman. A similar but smaller leukocytosis is also observed after the Olympic triathlon. As many as 84% of the subjects were above the laboratory reference limit for

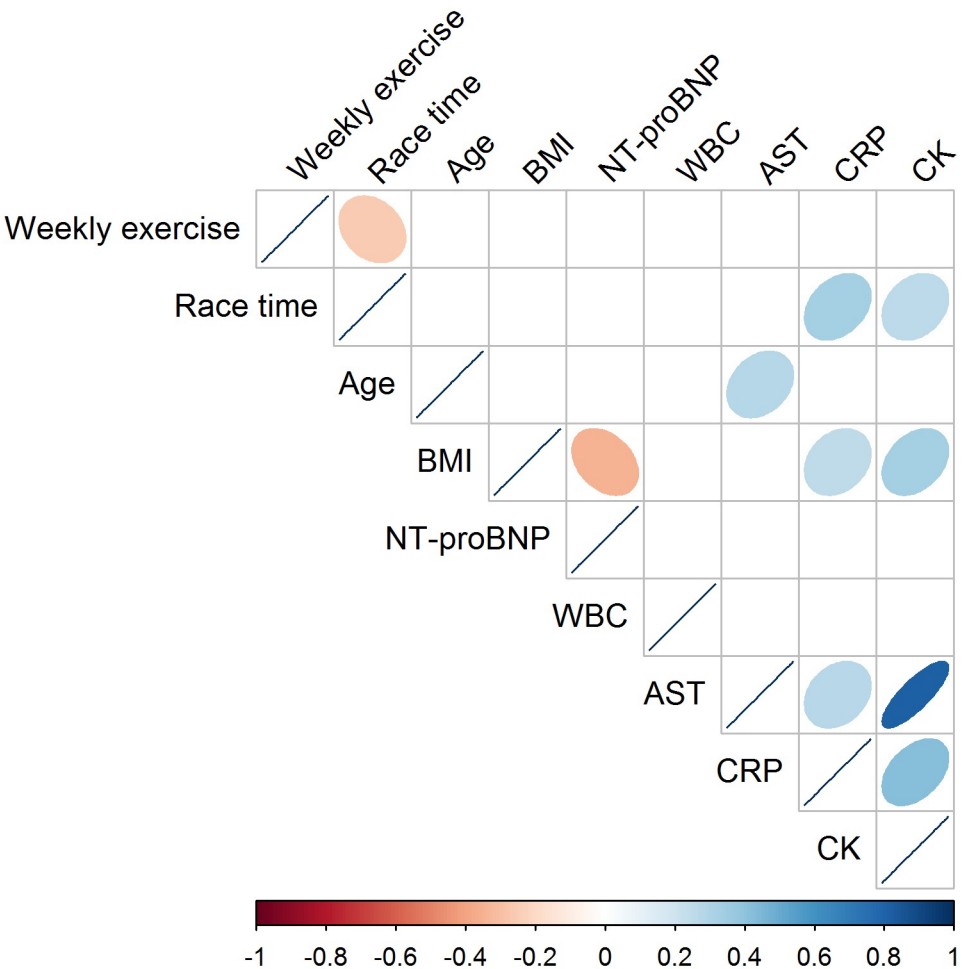

**Fig 1. Correlations.** Correlation matrix with Spearman correlations between the clinically significant elevations in blood samples: NT-proBNP, white blood cells (WBC) and Creatinine Kinase (CK) measured after finish; Aspartate Aminotransferase (AST) and C-reactive protein (CRP) measured the following day; reported weekly exercise; race time; age; and BMI. The correlation coefficient is visualised by *gradient colour* and *shape* to indicate negative correlation (red) and positive correlation (blue). Non-significant results are blank with the significance level set to $p < 0.05$.

WBC at the finish line after the Norseman, while only 27% of the subjects after the Olympic distance tested that high.

Exercise induced leukocytosis is a known physiological phenomenon described as early as at the end of the 19th century [27]. Circulatory catecholamine's are known to mobilise leukocytes from the spleen, lung and marginal zone in venules [28]. An increase in catecholamines during exercise is therefore believed to be of importance concerning exercise induced leukocytosis [29, 30]. However, exercise induced leukocytosis is also shown to occur during the infusion of non-selective beta-blockers [31]. Increased blood flow through tissue with pooled leucocytes is therefore believed to work together with humoral activation to mobilise leucocytes into the circulatory blood pool during exercise [30].

Exercise induced leukocytosis is shown to have a bimodal increase after shorter periods of exercise, with a partial initial increase at the finish, followed by a transient decrease after 15–30 min, before a final increase 90–120 minutes post exercise [32]. In a study of prolonged exercise during a 24 hour race, there was a gradually increasing leukocytosis for the first 16 hours,

**Table 2. Main results.**

| Variable | Race | Baseline | Finish | Day after |
|---|---|---|---|---|
| **Hb** | N | 14.8 [14.2–15.4] | 14.2 [13.5–14.9] * | 14.0 [13.2–14.5] *** |
| (g/100 mL) | O | 14.8 [14.4–15.2] | 14.4 [14.1–15.0] | 14.4 [14–14.6] |
| **WBC** | N | 4.0 [3.4–4.9] | 12.7 [11.1–15.9] *** | 7.8 [6–10] *** |
| (10^09 /L) | O | 3.2 [2.8–3.8] | 7.8 [6.7–9.6] *** | 5.3 [3.9–6.3] ** |
| **Thrombocytes** | N | 214 [197–243] | 246 [227–273] ** | 236 [208–255] |
| (10^09/L) | O | 191 [176–218] | 235 [196–274] | 202 [169–217] |
| **CRP** | N | 1 [1–1] | 8 [4–19] *** | 39 [27–56] *** |
| (mg/L) | O | 1 [1–1] | 1 [1–1] | 2 [1–3] *** |
| **Na$^+$** | N | 140 [139–141] | 140 [138–141] | 140 [139–141] |
| (mmol/L) | O | 143 [142–144] | 143 [142–144] | 141 [140–144] |
| **K$^+$** | N | 4.5 [4.3–4.7] | 4.3 [4.1–4.7] * | 4.2 [3.9–4.4] *** |
| (mmol/L) | O | 4.4 [4.2–4.6] | 4.7 [4.5–5.1] ** | 4.4 [4.3–4.7] |
| **Ca$^{2+}$** | N | 2.38 [2.32–2.43] | 2.45 [2.38–2.53] *** | 2.36 [2.30–2.42] |
| (mmol/L) | O | 2.44 [2.40–2.49] | 2.47 [2.44–2.54] | 2.38 [2.35–2.43] * |
| **Mg$^{2+}$** | N | 0.81 [0.77–0.85] | 0.90 [0.85–0.96] *** | 0.87 [0.84–0.92] *** |
| (mmol/L) | O | 0.79 [0.76–0.83] | 0.74 [0.68–0.76] ** | 0.82 [0.80–0.87] |
| **Creatinine** | N | 74 [66–82] | 94 [81–106] *** | 82 [73–91] *** |
| (umol/L) | O | 80 [73–82] | 98 [81–108] ** | 81 [74–83] |
| **AST** | N | 27 [24–32] | 99 [74–136] *** | 142 [99–191] *** |
| (U/L) | O | 22 [19–27] | 29 [23–34] | 31 [26–41] ** |
| **ALT** | N | 29 [24–35] | 42 [36–55] *** | 53 [42–65] *** |
| (U/L) | O | 27 [23–33] | 31 [24–34] | 29 [23–37] |
| **CK** | N | 158 [119–203] | 2450 [1620–3950] *** | 2910 [1650–4730] *** |
| (U/L) | O | 151 [97–225] | 303 [182–393] ** | 531 [314–670] *** |
| **NTproBNP** | N | 25 [20–49] | 576 [331–856] *** | 230 [145–380] *** |
| (ng/L) | O | 23 [20–29] | 77 [49–88] ** | 63 [45–103] ** |
| **TSH** | N | 1.6 [1.1–2.2] | 2.1 [1.1–3.2] | 1.5 [1.0–2.3] |
| (mU/L) | O | 2.6 [2.1–3.5] | 2.2 [1.7–2.5] | 1.3 [0.9–1.6] *** |
| **T3** | N | 4.9 [4.5–5.3] | 4.9 [4.3–5.5] | 4.7 [4.3–5.2] |
| (pmol/L) | O | 5.3 [5.0–5.8] | 5.1 [4.7–5.2] | 5.1 [4.7–5.3] |
| **T4** | N | 15.8 [14.5–17.1] | 18.6 [16.3–20.3] *** | 16.7 [15.6–18.2] * |
| (pmol/L) | O | 16.2 [15.6–18.2] | 16.3 [15.4–19.0] | 15.7 [14.8–17.2] |

Values are median [1. Quartile, 3. Quartile]. P-values were calculated with Wilcoxon Signed Rank Tests to assess changes from baseline to Finish and Day after measurements.

N, Norseman; O, Olympic triathlon; Hb, Hemoglobin; WBC, White Blood Cells; CRP, C-reactive protein; AST, Aspartate Aminotransferase; ALT, Alanine Aminotransferase; CK, Creatinine Kinase; NT-proBNP, N-terminal pro Brain Natriuretic Peptide; TSH, Thyroid Stimulating Hormone; T3, Triiodothyronine; T4, Thyroxine.

* p value < 0.05

** p value < 0.01

*** p value < 0.001

followed by a decrease throughout the rest of the race [33]. These different profiles of leukocytosis could explain the differences between the WBC seen after Norseman, with a race time of 10–20 hours, and that after the Olympic distance, with a race time of less than 3 hours for a majority of the athletes. The values observed after the Norseman are comparable to the values reported 3 hours after short and intense exercise protocols [18, 34], but higher than what is

**Table 3. Percentage athletes above reference values.**

| Variable | Race | Baseline | Finish | Day after |
|---|---|---|---|---|
| **Hb** | N | 3% | 0% | 0% |
| (> 17 g/100mL) | O | 7% | 7% | 0% |
| **WBC** | N | 0% | 84% | 22% |
| (> 10 x 10^9/L) | O | 0% | 27% | 0% |
| **Thrombocytes** | N | 0% | 0% | 0% |
| (> 600 x 10^9/L) | O | 0% | 0% | 0% |
| **CRP** | N | 1% | 65% | 98% |
| (> 5 mg/L) | O | 0% | 0% | 7% |
| **Na$^+$** | N | 0% | 0% | 0% |
| (> 145 mmol/L) | O | 7% | 20% | 7% |
| **K$^+$** | N | 6% | 7% | 2% |
| (> 5 mmol/L) | O | 7% | 27% | 7% |
| **Ca$^{2+}$** | N | 5% | 29% | 3% |
| (> 2.51 mmol/L) | O | 14% | 29% | 0% |
| **Mg$^{2+}$** | N | 2% | 32% | 19% |
| (> 0.94 mmol/L) | O | 0% | 0% | 0% |
| **Creatinine** | N | 0% | 31% | 7% |
| (> 105 umol/L)[f] | O | 0% | 33% | 0% |
| **AST** | N | 2% | 97% | 100% |
| (> 45 U/L)[f] | O | 0% | 13% | 27% |
| **ALT** | N | 2% | 22% | 34% |
| (> 70 U/L)[f] | O | 0% | 7% | 0% |
| **CK** | N | 6% | 99% | 99% |
| (> 400 U/L)[f] | O | 7% | 27% | 60% |
| **NT-proBNP** | N | 1% | 99% | 94% |
| (> 85 ng/L)[f] | O | 0% | 23% | 23% |
| **TSH** | N | 3% | 10% | 2% |
| (> 4 mU/L) | O | 7% | 0% | 0% |
| **T3** | N | 2% | 2% | 3% |
| (> 6.5 pmol/L) | O | 7% | 0% | 0% |
| **T4** | N | 0% | 2% | 2% |
| (> 23 pmol/L) | O | 0% | 7% | 0% |

N, Norseman; O, Olympic triathlon; Hb, Hemoglobin; WBC, White Blood Cells; CRP, C-reactive protein; AST, Aspartate Aminotransferase; ALT, Alanine Aminotransferase; CK, Creatinine Kinase; NT-proBNP, N-terminal pro Brain Natriuretic Peptide; TSH, Thyroid Stimulating Hormone; T3, Triiodothyronine; T4, Thyroxine.

[f] Different upper reference value for females. These are 90 umol/L for Creatinine, 35 U/L for AST, 45 U/L for ALT, 210 U/L for CK and 170 ng/L for NT-proBNP.

reported after ultra-distance marathons with a duration of 9–12 hours [17]. The different levels of leukocytosis between the Norseman and ultra-distance marathons could be related to differences in muscle activation [35]. In triathlons, the athletes use both the upper- and lower body more, compared to long distance running. Another possible explanation may be a difference in intensity, or the variation of intensity, during the races [36]. Further studies of long duration exercise events with registration of both intensity and leukocyte counts are needed to answer these questions.

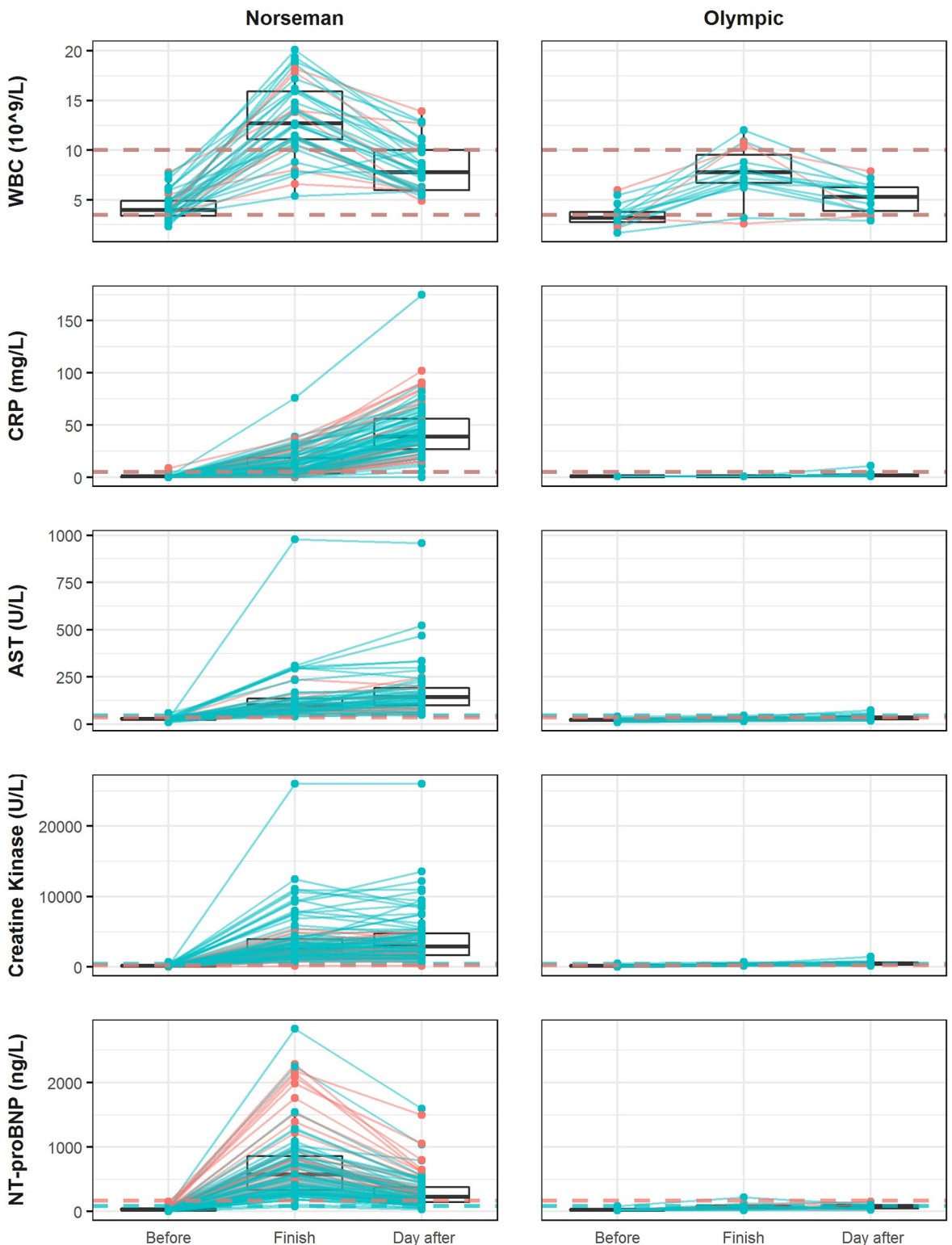

**Fig 2. Individual values.** Results for White blood cell count, CRP (C-reactive protein), AST (Aspartate Aminotransferase), Creatinine Kinase and NT-proBNP before the race, at the finish line and on the day after the Norseman (left) and Olympic (right) distances. Individual values are given with lines connecting each participant. Green points indicate male sex and red points indicate female sex. The green dotted line is a laboratory reference value for males; red is for females and brown indicates the same reference line for both sexes. The y-axis is identical for the Norseman and Olympic distances for each biochemical marker.

## CRP, AST and CK

CRP, AST and CK all showed clinically significant increases after the Norseman with, respectively, 65%, 97% and 99% of the participants found above reference values immediately after the race. CRP continued to increase the day after the race, with 98% of the participants reaching levels above reference values.

CRP is an acute phase protein originating from the liver [37] stimulated by IL-6 originating at sites of pathology [9, 38]. Pathology such as muscle damage thus elevates the circulatory pool of CRP to above reference values after 6 hours and peaks after 48 hours [37, 39, 40]. This explains the continued increase of CRP the day after the Norseman. After the Olympic distance, there was no increase directly after the race and almost no increase the following day. This is in line with other studies of shorter exercise that show little change in CRP [16, 41]. CRP after the Norseman was correlated to cell damage biomarkers (AST and CK), and both CRP and CK showed significant correlation with the race times in the Norseman. We therefore think that shorter events with exercise cause less damage and therefore a smaller stimulus to CRP production in the liver.

In contrast to this thesis, some recently published values of CRP during and after a 24 hour ultra-distance race were less then 10 mg/L [4] for all measurement times. However, the day following the Norseman we measured CRP to 39 [27 – 56] mg/L. This is comparable to recently published values from an Ironman triathlon [4]. It could be argued that there are differences in the fitness of the athletes in the 24 hour ultra-distance run and the Norseman. In our sample of athletes from the Norseman there are both professional triathletes and recreational triathletes, while the measurements in the 24 hour ultra-distance run had inclusion criteria stating that all subjects had to have participated in at least 5 marathons. A retrospect analysis of our data shows that only 2 out of 96 CRP values from the following day were beneath 10 mg/L and that the best triathletes also had higher CRP values. See Fig 3 for CRP values plotted against race times. This indicates that the form of exercise in an extreme triathlon, with use of the whole body and more varied intensity, is of importance to the CRP increase. This is possibly due to the associated cell damage in Ironman distance thritlons [9]. At the same time, the duration of exercise seems to be of importance. Further studies of both mechanical load, intensity and CRP are needed to better understand these differences in CRP values after the Norseman and 24 hour ultra-marathons.

Values for C-reactive protein (CRP) for all measured individuals. Red indicates female and blue indicates male. The black dashed line indicates 10 mg/L as referred to in the discussion.

Another interesting finding is that CRP was not correlated to the leukocytosis, but to the increase of CK and AST, which both are intracellular enzymes used as markers of cell damage [42]. This supports the idea that the exercise induced leukocytosis is a phenomenon that includes the mobilisation of white blood cells due to exercise induced humoral activation and increased blood flow rather than a phenomena due to cell damage [28, 43]. However we believe the CRP increase to be a response actual cell damage through IL-6 stimulation from activated leucocytes in damaged muscle tissue, elicited by prolonged and exhausting exercise [38].

## NT-proBNP

NT-proBNP is the N-terminal bi product of the prohormone for BNP, a cardiac hormone [44]. BNP causes diuresis, vasodilatation and decreased renin and aldosterone secretion and therefore reduces the load on the heart through reduced peripheral resistance and reduction in blood volume [45]. NT-proBNP is used as a surrogate for BNP due to longer half time in plasma [46]. BNP is believed to be released due to strain in myocytes [47]. Increased levels are

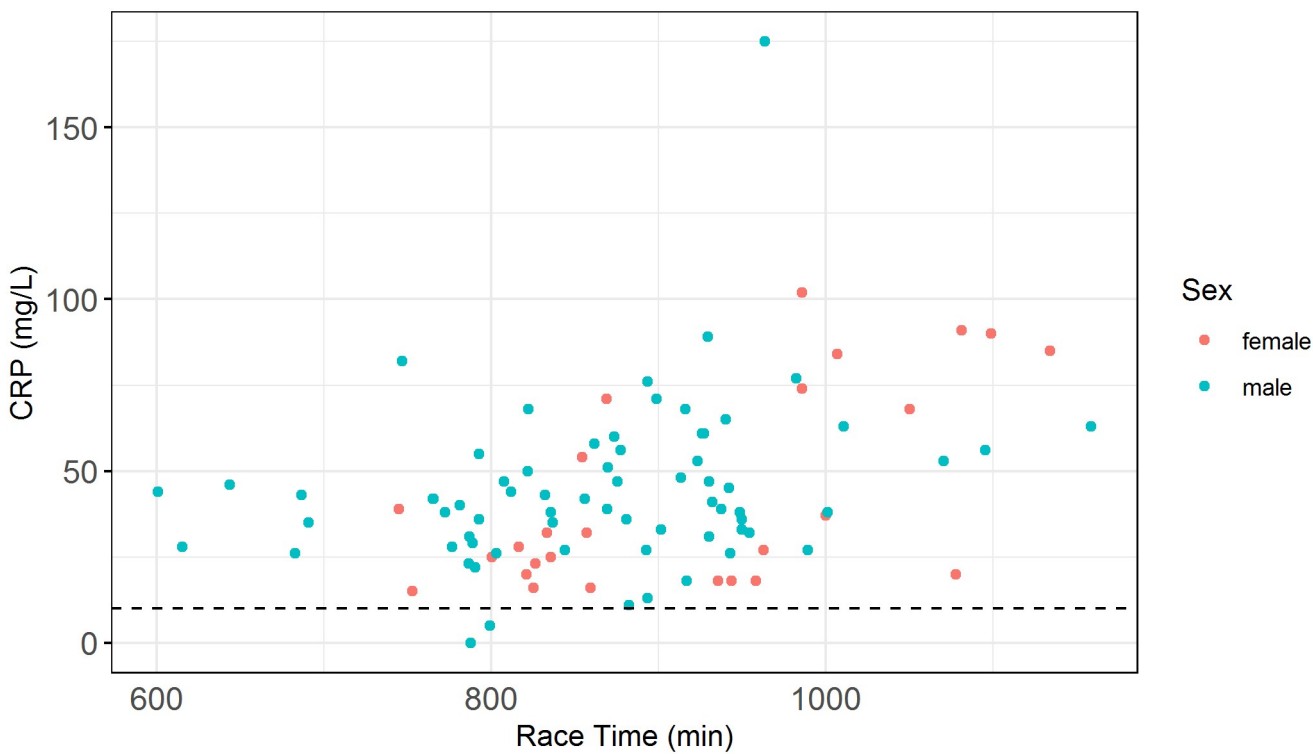

**Fig 3. Individual CRP values the day after the Norseman in relation to race times.**

also associated, to a lesser degree, with hypoxia [48–50]. Hormonal regulation is furthermore demonstrated in vitro with increased secretion in the presence of angiotensin II and reduced secretion in the presence of endothelin in rat myocytes [51].

NT-proBNP has become a biomarker for detecting and managing heart failure with prognostic value, where higher values is associated with increased mortality [26]. There is no consensus for cut-off values, but values of > 450 ng/L have been proposed as diagnostic for heart failure for a population < 50 years [52]. In our study the median value for NT-proBNP at the finish line after the Norseman was 576 ng/L, with results ranging from 76 ng/L to 2837 ng/L, with 61% of the participants having values above 450 ng/L. We observed that 99% of the participants had values above the laboratory reference limits (85 ng/L for men and 170 ng/L for women). There was a large and significant difference between the Norseman and the Olympic distance. The high values measured after the Norseman are in line with reported values after Ironman Kalmar in Sweden [8]. The results from the Norseman and Ironman Kalmar, indicate that long distance triathlons elicit considerably higher values than reported after both ultra-distance running [4] and long distance swimming [53]. This is probably due to the combination of long time exercise and high intensity causing repeated strain on the cardiac myocytes.

Interestingly, our correlation analysis showed a negative correlation to BMI, indicating that a higher BMI protects against a NT-proBNP rise. A study of NT-proBNP and cut off values for heart failure show decreased values in obese patients with heart failure [54]. These findings are in line with our findings after the Norseman, with lower values of NT-proBNP in subjects with larger BMI who have all been through the same race. The reason for lower values with higher BMI is not elucidated. However, one could speculate that the heart of athletes with larger body mass is relatively smaller to their blood volume causing a smaller concentration of NT-proBNP

in the blood [55, 56]. Another possibility is that the rate of secretion could be higher in larger subjects [55].

### Strengths and limitations

This study presents biochemical markers from a relatively large number of participants offering valuable reference values for clinicians treating patients and medical teams working in Ironman distance competitions. Differences between the genders have not been the main aim of this article. But expected values for several biomarkers differ between males and females as well as finish times, therefore supplemental information are provided with values for both sexes [13, 57–59]. Norseman is known for cold swim temperatures [60, 61]. A limitation of our study is that we have not studied the effects of swimming in cold water may affect the biomarkers [62]. Another limitation is variation in time between finish blood samples and blood samples from the day after the race. This was caused by the large variation in finish times and that all blood samples form the day after the race were collected at noon due to practical concerns.

### Conclusion

In conclusion, prolonged exercise events, such as Norseman, induces significant changes in biomarkers used in a clinical setting. Several are previously known to change during exercise, but this study describes the magnitude of increase in an Ironman triathlon competition, compared to a shorter competition, the Olympic distance triathlon. Of greatest clinical importance in the present study are the large increases in leukocytes, CRP, AST, CK and NT-proBNP after the Norseman. This is important to be aware of when athletes engaging in prolonged exercise competitions receive medical assistance or are hospitalised during or after an event like the Norseman. We publish our measured values after the Norseman as a guiding tool for expected physiological changes for clinicians treating patients and medical teams in prolonged exercise events.

### Supporting information

**S1 Table. Included samples.**
(DOCX)

**S2 Table. Results for only males.**
(DOCX)

**S3 Table. Results for only females.**
(DOCX)

**S4 Table. Differences between genders.**
(DOCX)

**S1 Raw data. CSV file containing raw data.** Age and race times are not included, to comply with the ethical approval of the study and Norwegian law, as this would make the subjects identifiable since results from the races are public.
(CSV)

### Acknowledgments

This study was conducted by the Norseman Research team. The Norseman Research team consists of scientists from serval institutions furthering the knowledge of extreme endurance exercise in cooperation with the Norseman Xtreme Triathlon and volunteering participants.

## Author Contributions

**Conceptualization:** Jørgen Melau, Maria Mathiasen, Jonny Hisdal.

**Data curation:** Christoffer Nyborg.

**Formal analysis:** Christoffer Nyborg.

**Funding acquisition:** Jonny Hisdal.

**Investigation:** Christoffer Nyborg, Jørgen Melau, Martin Bonnevie-Svendsen, Maria Mathiasen, Helene Støle Melsom, Jonny Hisdal.

**Methodology:** Christoffer Nyborg, Maria Mathiasen, Jonny Hisdal.

**Project administration:** Christoffer Nyborg, Jørgen Melau, Maria Mathiasen, Jonny Hisdal.

**Resources:** Jørgen Melau.

**Software:** Christoffer Nyborg.

**Supervision:** Martin Bonnevie-Svendsen, Jonny Hisdal.

**Validation:** Christoffer Nyborg, Jørgen Melau, Martin Bonnevie-Svendsen, Maria Mathiasen, Helene Støle Melsom, Andreas B. Storsve, Jonny Hisdal.

**Visualization:** Christoffer Nyborg.

**Writing – original draft:** Christoffer Nyborg.

**Writing – review & editing:** Christoffer Nyborg, Jørgen Melau, Martin Bonnevie-Svendsen, Maria Mathiasen, Helene Støle Melsom, Andreas B. Storsve, Jonny Hisdal.

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
