## [Decision Letter · Decision Letter 0]

8 Jun 2020

PONE-D-20-13546

High CRP, AST, CK, NT-proBNP and leukocytosis after the Norseman Extreme Triathlon (NXTRI)

PLOS ONE

Dear Dr. Nyborg,

Thank you for submitting your manuscript to PLOS ONE. After careful consideration, we feel that it has merit but does not fully meet PLOS ONE’s publication criteria as it currently stands. Therefore, we invite you to submit a revised version of the manuscript that addresses the points raised during the review process.

Please consider carefully all coments form the reviewers. It seems that ther is a general lck of proper references along the manuscript as well as spelling or grammatical mistakes. Furthermore, more details regarding methodology are required.

We look forward to receiving your revised manuscript.

Kind regards,

Pedro Tauler, Ph.D.

Academic Editor

PLOS ONE

Journal Requirements:

'The authors have declared that no competing interests exist.' 

We note that one or more of the authors are employed by a commercial company: Aker BioMarine Antarctic AS.

Additional Editor Comments (if provided):

Reviewers' comments:

Reviewer's Responses to Questions

**Comments to the Author**

1. Is the manuscript technically sound, and do the data support the conclusions?

Reviewer #1: Yes

Reviewer #2: Yes

Reviewer #3: Yes

2. Has the statistical analysis been performed appropriately and rigorously? 

Reviewer #1: Yes

Reviewer #2: Yes

Reviewer #3: Yes

3. Have the authors made all data underlying the findings in their manuscript fully available?

Reviewer #1: Yes

Reviewer #2: Yes

Reviewer #3: Yes

4. Is the manuscript presented in an intelligible fashion and written in standard English?

Reviewer #1: No

Reviewer #2: No

Reviewer #3: Yes

5. Review Comments to the Author

Reviewer #1: General comment

This is an interesting study on a popular sport, where the authors use correctly scientific methods. I would be happy to recommend it for publication once the authors address a few issues. My major concern is that the English and writing style must be improved throughout the text to meet the standards of PLOS ONE as in its current form some parts are hard to be understood.

Specific comments

1. Title: Adopt a more ‘neutral’ title, e.g. ‘the effect of … on…’ and delete (NXTRI)

2. l.19-24: Reduce this part in 2-3 lines.

3. l.25-30: Increase the information on participants and decrease the information on biomarkers (they are reported in l.31-40)

4. l.31-40: This part must be rewritten as it is hard to follow.

5. l.41: Add a conclusion in 2-3 lines with a practical perspective.

6. l.44: Major literature is missing (DOI: 10.4077/CJP.2016.BAE420).

7. l.49: Changes meters to m.

8. l.45-49: Add references to support the provided information.

9. l.50: The term ‘extreme exercise’ -here and throughout the text- must be revised as it is not precise. Is it prolonged? High-intensity? Extreme cold? Cold water? Large elevations? Eg the ‘prolonged strenuous’ used in l.59 is more precise.

10. l.50-58: Add 3-4 references.

11. l.61: Revise ‘extreme triathlon’.

12. l.61: The gap of the existed literature has not been identified correctly. It needs 3-4 sentences (with references) presenting what is known and what is missing.

13. l.67: Add hypotheses.

14. Introduction: Make this part more specific to the aims to the paper. All biomarkers presented in the methods and results must be introduced here.

15. l.72: When did they receive the email?

16. l.79: …noon of…

17. l.95-96: Delete references 6-13.

18. l.107: Delete ‘significant… 0.001’. Report all p values at 3 decimals, e.g. p=0.364, p<0.001…

19. l.108: Add effect size.

20. l.297: Add a paragraph showing limitations, strength and practical applications.

21. l. 297: Major literature is missing in the discussion, too (DOI 10.3389/fphys.2017.00638; 10.3389/fphys.2018.01959)

Reviewer #2: GENERAL

Thank you for sending me this interesting manuscript in which the authors have described changes in various biomarkers in response to the Norseman Extreme Triathlon. The aim of the study is to elucidate typical changes in these biomarkers, which might assist in risk stratification. The study is simple, with a straight-forward protocol. It provides useful data for Race Directors and medics who might be overseeing the event. I have no major issues with study or analyses, but more in the general write-up and reporting of Methods and Results. Important details are missing from the Methods, and sections need clarifying to help with interpreting the Results. For example, no data has been provided on the precise timescale of post-race blood samples, or in environmental conditions between Norseman and Oslo Olympic. These omissions undermine your interesting results. There are also numerous errors in language and in reporting of Results which may obscure the translation of the data. The study may deserve publication, but the manuscript needs to be considerably revised. Please see Specific comments.

SPECIFIC

Abstract

◦ Line 19-20. “Little is known about the physiological impact of such events on the human body”. I disagree with the statement; there are hundreds of studies looking at physiological responses to ultra- and extreme-endurance competition. Perhaps you could say ‘physiological impact of extreme triathlon’?

◦ Line 21. It seemed to me, from the Introduction and Discussion, that the main aim was to characterize the biomarker response, and that comparison to Olympic was a secondary aim for context. Consider re-ordering the abstract to reflect the primary and secondary aims.

◦ Line 24. Use of the word ‘injured’ here might not be appropriate; implies musculoskeletal injury.

◦ Line 25. Intravenous samples? Many of these biomarkers can be assessed with pin-prick capillary.

◦ Line 31. “Increased levels of clinical significance”. Just say ‘significant increases’.

◦ Line 33. “Clinically significant changes”. Changes could be increase or decrease, be specific.

◦ See my comment below in Results; the post-race absolute values need context. The absolute vealues presented here don't mean anything. Why not express the Results as percentage increase from baseline? This way you provide context against baseline values.

◦ Abstract needs a concluding statement. How might the data be used?

Keywords. These contain several misspelled words, e.g., ‘Extreeme’ and ‘Thriathlon’. Also, keywords are generally those not included in the title. Reconsider.

Title. The title is not descriptive, and also contains technical abbreviations. Please amend the title congruent with PLOS ONE submission guidelines (https://journals.plos.org/plosone/s/submission-guidelines). Titles should be “Specific, descriptive, concise, and comprehensible to readers outside the field. Avoid specialist abbreviations if possible”. This should have been highlighted by the journal before being sent for review.

Introduction

◦ Line 45. Given that the citations are non-academic resources, please caveat that Norseman has been rated by commercial/media/popular outlets as one of the world’s toughest triathlons.

◦ Line 46. Please use SI units for meters (m), here and throughout.

◦ Line 51. Please change ‘world’ to possessive world’s.

◦ Line 61. Others have already evaluated biomarkers in response to Ironman (Danielsson et al - PMID: 28609447; Carlsson et al - PMID: 27483401; Neubauer et al - PMID: 18548269). Why haven’t these studies been mentioned here?

Line 61. I think much of the rationale for the study is based on the idea that Norseman is in a different category of triathlon. For your paper to be unique, please expand upon why Norseman is more extreme than ‘normal’ Ironman distance tri.

Methods

◦ Line 71. It would serve you to mention that Norseman and Oslo Olympic were both contested in August (i.e., similar anticipated weather); I had to look this up as the data weren’t provided. What time of day were the starts? Do you have any data on the weather conditions during both races? This would support your argument that the races were comparable in all but distance/terrain, and would lend validity to your comparisons.

◦ Line 79. Please state an approximate timescale for the first post-race blood sample; i.e., ‘within 5 min of race completion’ or ‘within 1 h of race completion’.

◦ Line 79. Why ‘noon’ the following day? Surely you didn’t collect >100 samples at noon? Be specific. How many hours on average after the finish were samples collected? Include means and STDEV. When assessing biomarkers, and in a study designed to elucidate the time-course of recovery, these details are paramount.

◦ Blood Sampling. For clarity, I would recommend integrating the first and last paragraphs (the protocol) and then following with the second paragraph (the measures). Also, I’m assuming samples were taken from an antecubital vein, but please state in the text.

◦ Line 104. No problem with the stats, but I’m confused why you’ve used Bonferroni post-hocs only on AST, CRP, and CK. Did you not assess each variable at three timepoints at both events? Sorry if I’m missing something.

◦ Line 104. I think the stats section needs a little clarity on the ‘why’. Be specific with your reasons for performing a given test, spell it out. For example, ‘to assess biomarkers among the three time-points (pre-, post-, recovery), a Wilcoxon Signed Rank Test…’. Moreover, line 105 is the first mention of correlations. State why you’re doing these: ‘to assess for associations among the variables, a Spearman’s Rank…’. Assessing associations should be in the aims.

◦ Line 104. Please capitalize ‘Spearman’s’.

◦ Line 107. To avoid repetition, consider replacing the first sentence by saying that ‘critical alpha level was set as 0.05’.

Results

◦ Line 112. Please use ‘before, after, +24h’, or ‘pre, post, post-day’, or some equivalent, but avoid mixing your time-points as you’ve done here ‘pre, after, post-day’. There needs to be consistency here for clarity. In the tables you’ve used ‘before, finish, after’. Even this is vague. What about ‘Start, Finish, +24’?

◦ Line 114. These things happen. Is this necessary information, though? i.e., do you anticipate there being differences among the years? If not, consider removing this statement.

◦ Line 116. Please clarify what is meant by ‘in some instances, single vials were discarded’. For what reason?

◦ Line 117. ‘There are differences in the total number of analyses…’. This line seems to contradict line 112 where you mentioned that ‘113 subjects were included for analysis with full sets of samples’. I think what you mean is that 113 subjects provided blood samples start, finish, and +24, but that not every measure was assessed in each participant. Again, be specific. Why were all measures not available from all subjects?

◦ Table 1. I’m sure this will be adjusted in the journal formatting, but title should always be above tables and below figures.

◦ Table1. Why provide height to 2 decimal places, but performance times and ages to none?

◦ Table 1. Consider changing ‘weight’ to ‘mass’. Also, provide mass to 1 d.p.

◦ Table 1. Age SI unit is ‘y’

◦ Interesting in itself that males / females had comparable swim times.

◦ Line 124. WBC increased to 14.2 *109/L after the race. This is different to what’s reported in Table 2. What am I missing here?

◦ Results. The post-race absolute values are meaningless if not given in the context of the baseline values. E.g., CK up at 2450 U/L doesn’t tell me about the magnitude of the increase. Seeing as you provide the raw data in Table 2, why not express the Results as percentage increase from baseline? This way you are providing something different to what I can find myself in the table, but it also provides context against baseline values.

◦ Line 139. The correlations; are these the R values? If so, please clarify in the text.

◦ Line 140. ‘Between’ variables implies only two. ‘Among’ variables means more than two.

◦ Line 154. Non-significant correlations aren’t technically correlations.

◦ Table 2. Caption, please use ‘Wilcoxon’ instead on Wilcox.

◦ Fig 2. I appreciate your desire to show individual values, but with such a large sample there is far too much data on this graph. It is difficult to read. Either use a light grey line for individual data and a thick dark line for means, or just display the means and SD/IQ.

◦ Fig 2. Given the female sample is 1/3 of the male, do you expect there to be meaningful differences among the variables? Also, is a comparison between sexes an aim of the study? If not, why present the data as two cohorts on the graphs?

◦ Fig 1/correlations. I’ve not seen correlations shown this way before. I think it’s an interesting and unique way of showing the data. Does it really add to what’s been written in the text? At your discretion.

Discussion

◦ I would like to see some more comparisons of your data against values reported in other Ironman distance triathlons. Also, there is very little mention of the factors that make Norseman so unique; e.g., the cold-water temperatures, the tough ascents/descents (presumably greater muscle damage with downhill components in running). How might these factors influence your measured variables. This is important to distinguish your study from others looking at ‘normal’ Ironman.

◦ Line 200. ’Measurements of WBC show that extreme exercise is related leukocytosis’; please check, is the word ‘to’ missing here?

◦ Line 205. Needs a citation.

◦ Line 209. You have switched from US to UK spelling of leukocyte/leucocyte. Be consistent.

◦ Line 221. Could the difference be due to the relatively greater stress in ultra-marathon due to impact forces?

◦ Line 223. Do you mean that IL6 production elevates CRP, or is CRP produced by CRP in a positive feedback mechanism?

◦ Line 250. The 24 h race you cite comprised loops of a flat course, so I could believe that exercise intensity and/or strain was greater in Norseman, but I’m not sure about this for many/most 24-hour races performed on trails, mountains, in heat, etc. Could other conditions like the cold water have influenced the values?

◦ Line 263. Swap the word ‘thesis’ for ‘notion’ or ‘idea’.

◦ Line 289-290. Possibly the result of slower races times and, therefore, lower intensity? Although I see you found no correlation here between race time and BMI, which is unusual.

Reviewer #3: Line 22: the aim of the study was

Line 26: explain that n=98 was for 3 years

Lines 45-49: I suggest adding references such as Chin J Physiol. 2016 Oct 31;59(5):276-283. doi: 10.4077/CJP.2016.BAE420 or Springerplus. 2015 Sep 2;4:469. doi: 10.1186/s40064-015-1255-5

Line 50: add a reference

Lines 52-53: add a reference

Lines 60-61: add a reference

Line 61: the aim of the study was

Line 67: what is the hypothesis of your study?

Line 69: what were the criteria of inclusion/exclusion to the study?

Line 200: is related to

Lines 204-205: add a reference

Lines 206-207: add a reference

Lines 207-208: add a reference

Lines 220-221: add a reference

Lines 221-222: add a reference

Lines 222-223: add a reference

Line 232: add a reference

Lines 232-233: add a reference

Lines 250-252: add a reference

Lines 252-253: add a reference

Line 261: was not related to

Line 261-262: add a reference

Lines 262-265: add a reference

Lines 265-266: add a reference

Line 269: add a reference

Lines 269-270: add a reference

Lines 270-271: add a reference

Lines 271-271: add a reference

Lines 284-285: incomplete sentence

Line 289: showed a negative

Line 289: indicating that a higher

Lines 293-295: add a reference

Lines 295-296: add a reference

6. PLOS authors have the option to publish the peer review history of their article (what does this mean?). If published, this will include your full peer review and any attached files.

Reviewer #1: No

Reviewer #2: No

Reviewer #3: No

---

## [Author Response · Author response to Decision Letter 0]

17 Aug 2020

Response to Reviewer #1

• We have changed to a more ‘neutral’ title.

• The abstract is revised based on your and the other reviewers input.

• Thanks for relevant major literature. This is included in the introduction and discussion.

• Changes meters to m in the characteristics table.

• References are added were requested.

• The term “extreme exercise” is revised to “prolonged exercise” as suggested.

• The need for reference values for medical teams is the main gap in literature and is described.

• The participants received e-mails 1-2 months prior to the events. This is included in the methods chapter.

• Hypotheses and aim has been clarified in the introduction.

• The introduction has been simplified and made more specific.

• Sugested grammatical changes have been altered

• A paragraph with strengths and limitations has been added. Practical applications have been clarified in the introduction and conclusion.

Response to Reviewer #2

• The concerns about time between measurements and the issue with swimming in cold water are addressed in a new paragraph with strengths and limitations.

• We agree that much is known about the physiological impact of prolonged exercise and have reformulated our introduction. We have also included the major literature you have made us aware of. 

• The word “injury” is removed due to revision of the abstract. The word accident is used in the introduction, as this is a major concern in large scale Ironman races as Norseman.

• We did intravenous blood samples form the antecubal vein. This is clarified in the method section.

• We have used the word “changes” as some of the measured electrolytes could potentially also shown decreased values. However all clinically significant changes that we found where increased values.

• We have chosen to keep the absolute values in the results chapter. The reasons are that the aim of our article is to provide reference values for clinicians and therefore the absolute value is our primary goal. As well as all of these biomarkers and their values should be familiar for medical personnel and it is absolute values that are used in the clinic, not relative change. However the relative change can easily bed deduced from the tables and the raw data if warranted.

• The practical usage of the data is addressed.

• SI units have been corrected. We chose to change the scale to cm due to a rounding-off issue addressed in the review process.

• We did not find any major changes between other papers describing biomarkers after Ironman triathlons.

• The statistics section is simplified and revised according to input from you and other reviewers.

• We changed to utilize “Baseline”, “Finish” and “Day After” throughout the article to simplify the reading of the time points.

• It was R-values in the correlations; this is clarified in the statistics section.

• You spotted a honest mistake about the value of the WBC at finish line. By mistake it was switched with the Hb value. This is corrected.

• We decided to keep individual connected values in Figure 2. We also would like to keep separation of the genders by color. We don’t believe gender differences to be a major concern in the article, but there are some differences. This is addressed in the new strengths and limitation section and made available to study through the supplemental information provided. There are also different reference values between the sexes for some of the biomarkers. Therefore we would like to fully visualize the data for readers familiar with the biomarkers and these gender differences (e.g. NT-proBNP) so they can be evaluated. Connected individual values give both the size of measure, spread and the direction of change (the main parameter in most statistical tests). So we believe this figure gives valuable information to certain readers. 

• It can often be hard to understand the clustered connection in multiple correlations. We therefore believe the visualized matrix give valuable insight. While the text give exact coefficients and p-values.

• We have uniformed the use of leukocyte/leucocyte.

• The text concerning IL-6 and CRP is clarified. IL-6 is produced by leucocytes at site of pathology. IL-6 then stimulates CRP production in the liver.

Response to Reviewer #3

• Major literature that you provided is now included.

• Hypothesis and aim is clarified.

• The inclusion criteria are clarified. We had no exclusion criteria.

• References are provided were warranted.

---

## [Decision Letter · Decision Letter 1]

1 Sep 2020

Biochemical markers after the Norseman Extreme Triathlon

PONE-D-20-13546R1

Dear Dr. Nyborg,

We’re pleased to inform you that your manuscript has been judged scientifically suitable for publication and will be formally accepted for publication once it meets all outstanding technical requirements.

Kind regards,

Pedro Tauler, Ph.D.

Academic Editor

PLOS ONE

Additional Editor Comments (optional):

Reviewers' comments:

Reviewer's Responses to Questions

**Comments to the Author**

1. If the authors have adequately addressed your comments raised in a previous round of review and you feel that this manuscript is now acceptable for publication, you may indicate that here to bypass the “Comments to the Author” section, enter your conflict of interest statement in the “Confidential to Editor” section, and submit your "Accept" recommendation.

Reviewer #1: All comments have been addressed

Reviewer #3: All comments have been addressed

2. Is the manuscript technically sound, and do the data support the conclusions?

Reviewer #1: Yes

Reviewer #3: Yes

3. Has the statistical analysis been performed appropriately and rigorously? 

Reviewer #1: Yes

Reviewer #3: Yes

4. Have the authors made all data underlying the findings in their manuscript fully available?

Reviewer #1: Yes

Reviewer #3: Yes

5. Is the manuscript presented in an intelligible fashion and written in standard English?

Reviewer #1: Yes

Reviewer #3: Yes

6. Review Comments to the Author

Reviewer #1: no commentno commentno commentno commentno commentno commentno commentno commentno commentno comment

Reviewer #3: The authors have addresssed all my comments and improved the manuscript accordingly which is now ready to be accepted for publication

7. PLOS authors have the option to publish the peer review history of their article (what does this mean?). If published, this will include your full peer review and any attached files.

Reviewer #1: No

Reviewer #3: No

---

## [Editor Report · Acceptance letter]

10 Sep 2020

PONE-D-20-13546R1

Biochemical markers after the Norseman Extreme Triathlon

Dear Dr. Nyborg:

I'm pleased to inform you that your manuscript has been deemed suitable for publication in PLOS ONE. Congratulations! Your manuscript is now with our production department.

Kind regards,

on behalf of

Dr. Pedro Tauler 

Academic Editor

PLOS ONE